# Hierarchical Autoregressive Transformers: Combining Byte- and Word-Level Processing for Robust, Adaptable Language Models

**Pit Neitemeier, Björn Deiseroth, Constantin Eichenberg & Lukas Balles**
Aleph Alpha Research
Heidelberg, Germany
`<firstname>.<lastname>@aleph-alpha-ip.ai`

## Abstract

Tokenization is a fundamental step in natural language processing, breaking text into units that computational models can process. While learned subword tokenizers have become the de-facto standard, they present challenges such as large vocabularies, limited adaptability to new domains or languages, and sensitivity to spelling errors and variations. To overcome these limitations, we investigate a hierarchical architecture for autoregressive language modelling that combines character-level and word-level processing. It employs a lightweight character-level encoder to convert character sequences into word embeddings, which are then processed by a word-level backbone model and decoded back into characters via a compact character-level decoder. This method retains the sequence compression benefits of word-level tokenization without relying on a rigid, predefined vocabulary. We demonstrate, at scales up to 7 billion parameters, that hierarchical transformers match the downstream task performance of subword-tokenizer-based models while exhibiting significantly greater robustness to input perturbations. Additionally, during continued pretraining on an out-of-domain language, our model trains almost twice as fast, achieves superior performance on the target language, and retains more of its previously learned knowledge. Hierarchical transformers pave the way for NLP systems that are more robust, flexible, and generalizable across languages and domains.

## 1 Introduction

Tokenization plays a fundamental role in natural language processing (NLP) as it breaks down text into units that computational models can process. Two fundamental approaches are character-level and word-level tokenization. While character-level tokenization uses the "atomic" units of text and enjoys a small vocabulary size, it leads to long sequences with a high computational and memory cost. Conversely, word-level tokenization leads to short sequences but suffers from extremely large vocabulary sizes and the inability to process out-of-vocabulary words.

Subword tokenization has emerged as a compromise between these two extremes and has become the standard. Common subword tokenizers are trained—separately from the model—on a reference corpus of text. For example, Byte Pair Encoding (BPE; Gage, 1994; Sennrich et al., 2016) builds a vocabulary starting from individual bytes and iteratively merging adjacent pairs of tokens that occur most frequently in the corpus until the desired vocabulary size is reached. The resulting subword vocabulary leads to good sequence length compression on the reference corpus, while maintaining the ability to handle out-of-vocabulary words using a byte fallback.

However, subword tokenizers come with several downsides. First, contemporary models routinely use vocabulary sizes in the hundreds of thousands, making the corresponding embedding matrices and output heads extremely large. For instance, for the 8B model of the Llama-3 family (Dubey et al., 2024), with a vocabulary size of 128k, embedding and head account for roughly 13% of the model's parameter footprint. Secondly, the tokenizer is fitted in a separate step and not included in the end-to-end learning process of the model. This becomes problematic when a pretrained model

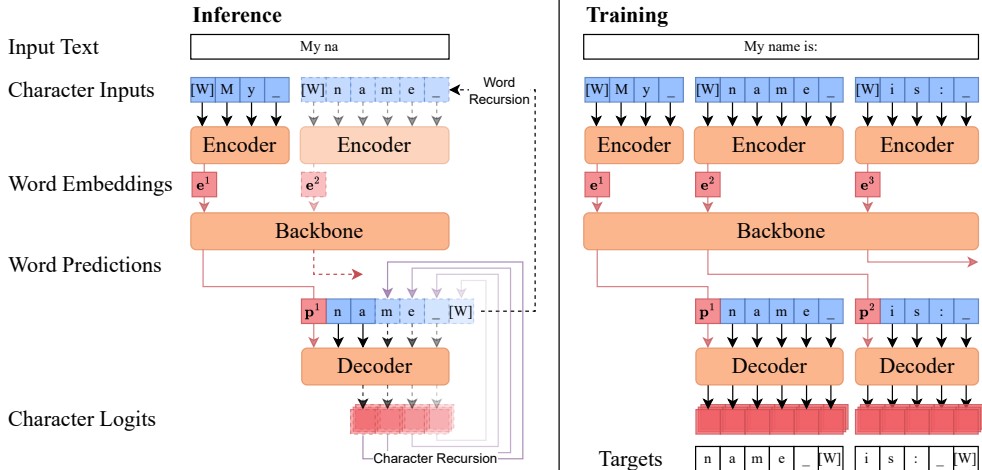

Figure 1: Schematic of the proposed hierarchical architecture. The input text is first split into words, with each word prepended by a special token [W]. These words are passed through the encoder and the activation at the position of the [W] token is selected as the word embedding $\mathbf{e}^i$. The sequence of word embeddings $\mathbf{e}^i$ is then processed by the backbone to produce abstract *predictive word embeddings* $\mathbf{p}^i$. The decoder then maps $\mathbf{p}^i$ to probabilites for the characters of the *next* word. During inference, given a text with a partial word, complete words are processed by the encoder and backbone. The decoder then *recursively* completes the remaining characters of the incomplete word and the completed word enters the encoder in a *word recursion*.

is applied to, or finetuned on text from different domains or languages to which the tokenizer is not attuned (see, e.g., Ali et al., 2024; Petrov et al., 2023; Deiseroth et al., 2024). Finally, spelling mistakes or variations can lead to drastically different token sequences for semantically close inputs and thereby degrade model performance.

To address these shortcomings, we investigate a hierarchical architecture as shown in Figure 1, which combines character-level and word-level processing. We first split the text into words. The characters of each word are processed by a small character-level encoder module, which maps them to a word embedding. The resulting sequence of word embeddings is then processed by a larger backbone model. The outputs of the backbone are treated as abstract "predictive" word embeddings and are decoded back to characters by another small character-level decoder module. As we will demonstrate, the character-level modules can be kept very small, having fewer parameters than the token-level embedding and head they replace. Encoder, backbone and decoder are transformer models and the entire system can be trained end-to-end, without the need for a fixed, trained tokenizer.

Our core contributions are:

- We revisit, refine and thoroughly investigate a **hierarchical architecture** for autoregressive language modelling that combines character- and word-level processing. Its design eliminates the need for a fixed word or subword-level vocabulary, and does not require separate tokenizer training. Through a careful computational cost analysis and comprehensive architecture sweeps, we identify optimal hierarchical model configurations across various compute budgets.

- We conduct **extensive experiments**, comparing our proposed model against state-of-the-art subword tokenizer-based models and a competing hierarchical approach in compute-matched experiments on identical data. We demonstrate that our approach scales effectively to the 7B scale, consistently matching the performance of the baseline models.

- We demonstrate that our proposed model is significantly **more robust** to perturbations of its input.

- Finally, we show that our model enjoys **superior finetunability** on out-of-distribution data, such as new languages or domains, outperforming tokenizer-based architectures which struggle with adaption to new vocabularies. In particular, during continued pretraining on an unseen language, our model achieves superior performance on the target language, retains more of its previously acquired knowledge, while training almost twice as fast.

The remainder of this paper is organised as follows. Section 2 introduces our hierarchical architecture, including a detailed computational cost analysis. Related work is discussed in Section 3. In Section 4, we present our experimental setup and results. Section 5 concludes the paper with final remarks and potential directions for future research.

## 2 HIERARCHICAL AUTOREGRESSIVE TRANSFORMERS

We now introduce our hierarchical architecture, which is a refinement and simplification of similar architectures proposed in prior works, see Section 3. Our approach relies on a splitting rule that partitions text into sequences of words. Specifically, we use UTF-8 bytes[1] as our base alphabet, consisting of $V_B = 256$ distinct values[2], and split the text at Unicode whitespace characters, which are appended to the previous word. A text can then be represented as $(w^1, \ldots, w^L)$ with $w^i \in [V_B]^{\ell_i}$ being a word of length $\ell_i$. Importantly, the splitting rule is the only non-trainable processing step in our method. We argue that, for natural text in (alphabetic) languages, whitespace splitting is adequate. However, our hierarchical architecture is agnostic to the type of splitting rule, allowing for alternatives that may be more appropriate for different languages or domains.

### 2.1 HIERARCHICAL ARCHITECTURE

Our architecture consists of three main components:

- An **encoder** $E \colon \mathbb{R}^{\mathbb{N} \times d} \to \mathbb{R}^{\mathbb{N} \times d}$, a bidirectional transformer operating on the character embeddings within each word.
- A **backbone** $B \colon \mathbb{R}^{\mathbb{N} \times D} \to \mathbb{R}^{\mathbb{N} \times D}$, a causal transformer operating on word embeddings.
- A **decoder** $D \colon \mathbb{R}^{\mathbb{N} \times d} \to \mathbb{R}^{\mathbb{N} \times V_B}$, a causal transformer with a language modelling head, operating on character level and outputting next-character prediction logits.

In addition, we have a character embedding $C \colon [V_B] \to \mathbb{R}^d$ and two projection matrices $\mathbf{W}_E \in \mathbb{R}^{D \times d}$ and $\mathbf{W}_D \in \mathbb{R}^{d \times D}$ mapping between the (smaller) character-level dimension $d$ and the (larger) word-level dimension $D$.

A document $(w^1, \ldots, w^L)$ is processed by the model as explained in the following and depicted in Figure 1. Following Devlin et al. (2019), each word is prepended with a special token [W] and its characters are embedded via $C$,

$$\mathbf{x}_j^i = C(w_j^{(i)}) \in \mathbb{R}^d, \quad \mathbf{x}_{[W]} = C([W]) \in \mathbb{R}^d. \tag{1}$$

Then, it is passed through the encoder. The output corresponding to the [W] token, i.e., the first entry in the sequence dimension, is selected as a *word embedding*:

$$\mathbf{e}^i = \left[ E(\mathbf{x}_{[W]}, \mathbf{x}_1^i, \ldots, \mathbf{x}_{\ell_i}^i) \right]_1 \in \mathbb{R}^d. \tag{2}$$

The resulting sequence of word embeddings is projected to the backbone dimension, passed through the backbone and projected back to the decoder dimension:

$$\tilde{\mathbf{e}}^i = \mathbf{W}_E \mathbf{e}^i \in \mathbb{R}^D, \quad \tilde{\mathbf{p}}^i = \left[ B(\tilde{\mathbf{e}}^1, \ldots, \tilde{\mathbf{e}}^L) \right]_i \in \mathbb{R}^D, \quad \mathbf{p}^i = \mathbf{W}_D \tilde{\mathbf{p}}^i \in \mathbb{R}^d. \tag{3}$$

The output for the $i$-th word, $\mathbf{p}^i$, is treated as a *predictive word embedding*, to be decoded into a sequence of characters matchting the *next* word. To that end, during training, we concatenate $\mathbf{p}^i$ with the character embeddings of the next word and map them through the decoder, resulting in a sequence of next-character prediction logits:

$$\mathbf{l}_j^i = \left[ D(\mathbf{p}^i, \mathbf{x}_1^{i+1}, \ldots, \mathbf{x}_{\ell_{i+1}}^{i+1}) \right]_j \in \mathbb{R}^{V_B}. \tag{4}$$

We train on the loss

$$\sum_{i=1}^{L} \sum_{j=1}^{\ell_{i+1}+1} \mathcal{L}\left( \mathbf{l}_j^i, w_j^{i+1} \right), \tag{5}$$

where $\mathcal{L}$ denotes character-level cross-entropy loss and we set the final prediction target of each word to $w_{\ell_{i+1}+1}^{i+1} = [W]$, indicating the end of the word.

---

[1]In the following, we use the terms character and byte interchangeably, but want to highlight that our approach is not specific to byte-level modelling.

[2]UTF8 has unused byte values, which we exploit for our special tokens, keeping the vocab size at 256.

## 2.2 INFERENCE

Inference in our hierarchical architecture proceeds in a nested loop, as shown on the left side of Figure 1. To generate a new word, we pass the context sequence through encoder and backbone to produce a predictive next-word embedding. To materialize the next word in characters, we run an autoregressive loop of the decoder module. When a word is completed, as indicated by the prediction of a [W] token, it is appended to the input and the process is repeated.

## 2.3 COMPUTATIONAL AND MEMORY COST

We now discuss the computational cost and memory footprint of our proposed architecture, contrasting it with a baseline model using a subword tokenizer and a corresponding embedding matrix and language modelling head. We assume the baseline model uses $P_{\text{baseline}}^{\text{backbone}}$ parameters in its backbone and $P_{\text{baseline}}^{\text{head}}$ in its embedding and head. For the hierarchical model, we assume $P_{\text{hierarchical}}^{\text{backbone}}$ parameters in the backbone and $P_{\text{hierarchical}}^{\text{char}}$ in each of the character-level modules.

**Computational Cost.** We present a simplified computational cost analysis under the assumption that the cost of a forward-backward pass through a transformer is proportional to $SP$, where $S$ is the sequence length and $P$ is the number of *non-embedding* parameters. This is a standard simplification based on the observation that feed-forward FLOPs dominate attention FLOPs for typical settings. An exact comparison, factoring in attention FLOPs, may be found in Appendix A.2.

Consider a document of $S$ characters. The computational cost of the two models depends heavily on the length of the sequence passed through the backbone, i.e., the number of words $S_{\text{W}}$ and tokens $S_{\text{T}}$ contained in the document, respectively. The baseline model processes a sequence of length $S_{\text{T}}$, passing it through the backbone and the output head, incurring a total cost of

$$C_{\text{baseline}} = S_{\text{T}} P_{\text{baseline}}^{\text{backbone}} + S_{\text{T}} P_{\text{baseline}}^{\text{head}}. \tag{6}$$

In the hierarchical architecture, the backbone processes a sequence of length $S_{\text{W}}$. Additionally, the two character-level models each process the sequence of length $S$ plus an additional $S_{\text{W}}$ word separator tokens for a total cost of

$$C_{\text{hierarchical}} = S_{\text{W}} P_{\text{hierarchical}}^{\text{backbone}} + 2(S + S_{\text{W}}) P_{\text{hierarchical}}^{\text{char}}. \tag{7}$$

Our experiments will show that small en-/decoder modules are viable. For instance, in our 3B-scale experiment, the computational cost of our two character-level modules roughly equals that of the language modelling head of the baseline. Additionally, words represent a coarser unit than subwords; our pretraining dataset exhibits a ratio of $S_{\text{W}} \approx 0.69 S_{\text{T}}$, see Figure 2 (left). Consequently, for a given cost budget, our hierarchical model will be able to operate with a larger backbone. We will revisit this for our compute-matched experiments in Section 4.

**Memory.** For compute-matched models, a hierarchical architecture will have a larger parameter footprint, see Table 2. In terms of activation memory, we have to distinguish between training and inference. During training, all activations have to be stored. Our hierarchical model shrinks the number of activations in the backbone by a factor of $S_{\text{W}}/S_{\text{T}}$ and avoids large logit tensors, but operates with a slightly larger backbone and needs to store additional activations in the character-level modules. Since the exact size of the activation memory depends on the caching strategy of the auto-differentiation framework, we forego a detailed comparison.

**Inference.** Using KV caching (Pope et al., 2023), it is possible to achieve near parity of FLOPs during training and inference. When comparing compute-matched models, we argue that a hierarchical architecture has a modest advantage in wall-clock time performance, see Appendix A.4. Additionally we propose a scheme to further optimise inference performance using cached word embeddings. The relevant quantity for memory consumption at inference time is the size of the KV cache, which can be a bottleneck in high-throughput inference systems. Here, hierarchical models lead to a smaller memory load, as detailed in Appendix A.5.

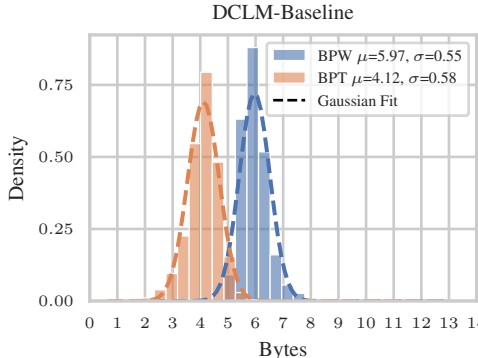 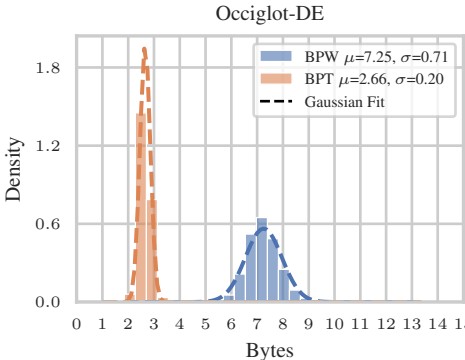

Figure 2: Bytes per word (BPW) and bytes per token (BPT) statistics, showing that words are a coarser unit than subword tokens. The tokenizer has been fitted to the DCLM-baseline dataset (left). On a dataset to which the tokenizer is not attuned, such as the German Occiglot dataset (right), the tokenizer "fragments" and BPT drops significantly.

## 3 RELATED WORK

The *granularity* of computational models for natural language has been a prominent question in NLP research since its inception (see, e.g., a review by Mielke et al., 2021). For many years, the question of character- vs word-level modelling has dominated the discussion, until subword tokenization methods, such as Byte-Pair Encoding (BPE, Sennrich et al., 2016), have emerged as a middle ground, balancing the flexibility of characters with the semantic coherence of words. Over time, subword tokenizers have become the dominant approach, gaining widespread adoption. The current landscape of NLP largely treats subword tokens as fundamental, indivisible units that are determined through a preprocessing step prior to model training.

Some prior works have augmented word or subword token embeddings with character-level information (e.g., Ma et al., 2020; El Boukkouri et al., 2020; Aguilar et al., 2021). More recently, the T-FREE method (Deiseroth et al., 2024) operates at the word level while incorporating character information through specialized embedding and output layers. These approaches have either not tackled generative modelling or still require a fixed vocabulary for generation.

Another line of prior work aims to enable character- or byte-level language modelling at contemporary scales. While some attempts have been made to train purely character-level transformers (Al-Rfou et al., 2019; Choe et al., 2019; Xue et al., 2022) these have ultimately not kept pace with subword-level models. Moving closer to the present work, some authors have presented "hybrid" approaches, e.g., using downsampling mechanisms (Charformer; Tay et al., 2022), or cross-attention with a latent "bottleneck" sequence (Perceiver AR; Hawthorne et al., 2022) to internally condense the large sequence lengths generated by character-level models. In the following, we describe in detail the most closely-related works.

Sun et al. (2023) use a hierarchical character-word architecture for BERT-style masked language modelling. As input to the decoder, the authors concatenate the backbone output for the $i$-th word with the per-character encoder outputs for the *same* word. We devise a generative variant of this architecture. We introduce a shift by one, concatenating with embeddings of the *next* word and use the raw character embeddings rather than the encoder outputs, since the bidirectional encoder would otherwise "leak" information about future characters. Our work also substantially scales up this modelling paradigm, experimenting with models up to the 7B parameter scale, compared to Sun et al. (2023) who use models around the 100M scale.

The MegaByte architecture (Yu et al., 2023) uses a backbone-decoder architecture for generative modelling. Instead of splitting byte sequences into words, MegaByte chunks it into fixed-size patches of subsequent bytes. Their architecture does not use an encoder module; the input to the backbone is simply the concatenation of the embeddings of the bytes within a patch. Note that this restricts the architecture to the use of fixed-size patches. At the decoder, the backbone output is

*added* to the byte sequences, which is prepended with a padding token. Yu et al. (2023) experiment with language models up to 320M parameters for the baseline model. We compare our approach experimentally to MegaByte below.

Thawani et al. (2023) propose the hierarchical architecture most closely related to the present work. A notable difference is that they prepend not one but four [W] tokens to each word in order to increase model capacity when going from encoder to backbone. This incurs drastically higher cost in en- and decoder compared to our approach of incrasing the hidden dimension. Thawani et al. (2023) experiment with models up to 77M parameters on datasets with fewer than 10M characters and a context window of only 192 characters (or the token equivalent thereof). Unfortunately, their experiments are not compute-matched and, by our calculation, assign 4x more compute to the hierarchical architecture compared to the baseline.

Finally, in work that appeared concurrently with the preparation of the present paper, Slagle (2024) propose a byte-level model that applies additional Transformer layers to a subset of the input bytes. They investigate both a fixed-size spacing as well as a split rule that marks only certain bytes as "split bytes", including whitespaces and punctuation. The byte-level layers are not restricted to individual words/chunks and instead use sliding window attention, precluding inference-time performance improvement via caching (Appendix A.4). Experiments are compute-matched and scaled up to the 1B models trained on 80B bytes and do not include downstream evaluations. None of the above papers investigate robustness or finetunability.

## 4 EXPERIMENTS

We proceed with an experimental investigation of the proposed method.

**Models.**   All models are based on the Llama architecture (Touvron et al., 2023) with a fixed attention head size of 128. For the baseline model and the backbone of our hierarchical architecture, we use Llama's default 1:1 "aspect ratio", i.e., the number of heads equals the number of layers. The design of the character-level encoder and decoder modules is discussed in Section 4.1. The baseline model uses a BPE tokenizer with a vocabulary size of 64k, fitted on our pretraining data.

**Data.**   We perform our main experiments using the DCLM-Baseline dataset (Li et al., 2024), which is a curated English-only pretraining dataset. Hyperparameter sweeps and ablations conducted early on in the project used the well-established Fineweb dataset (Penedo et al., 2024). For our continued pretraining experiment, we use the German portion of the Occiglot Fineweb v0.5 dataset (Brack et al., 2024). Dataloading is handled on a byte basis to guarantee that both models get to see the exact same data during training. We enforce a maximum document length of $16,384$ bytes, corresponding to roughly 4k tokens or 2.7k words. We load batches of $1024 \cdot 16\,384$ bytes, packing together documents of varying lengths with appropriate attention mask reset. In our pretraining experiments, we train for 72k steps, which comes down to a total training set size of roughly 1.2 trillion bytes.

**Hyperparameters.**   We use the AdamW optimiser (Loshchilov & Hutter, 2019) using $\beta_1 = 0.9$, $\beta_2 = 0.95$, $\varepsilon = 10^{-8}$ and weight decay coefficient $\lambda = 0.1$. The learning rate is warmed up over 500 steps followed by a cosine decay to $10\%$ of its peak value. We did not tune learning rates individually for each model but instead opted to use a well-established heuristic, scaling the learning rate inversely proportional to model width, see Appendix B.1.

**Eval Metrics.**   We focus on downstream evaluations as the primary metric for comparison, see Appendix B.4 for a description of our eval suite. In addition, we report a more immediate metric for pretraining performance. Since we compare models making byte-level and subword-level predictions, this requires some extra care. We use accuracy aggregated at the word level, as explained in Appendix B.2.

### 4.1 HIERARCHICAL ARCHITECTURE SWEEP

Our hierarchical model consists of encoder, backbone, and decoder. Given a total compute budget, we now conduct a series of experiments to determine optimal sizes (number of heads and layers) of these three modules. These experiments used the Fineweb dataset at a budget of 14.4k steps.

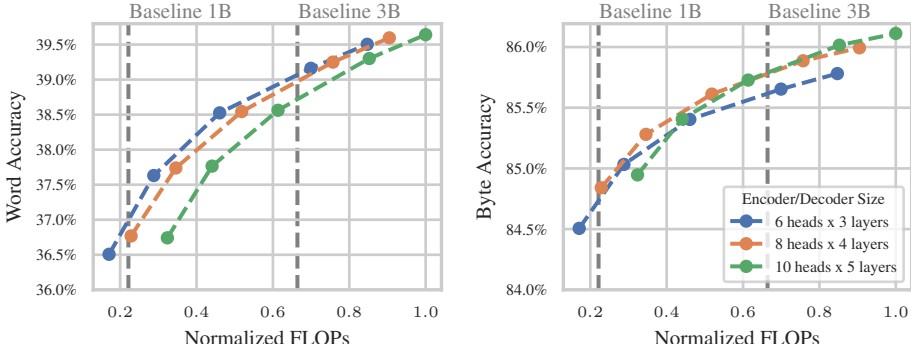

Figure 3: Word and byte accuracy for hierarchical models with different encoder/decoder sizes. Each candidate encoder/decoder size has been trained with backbone sizes ranging from $L_b = H_b = 16$ to $L_b = H_b = 30$. The horizontal axis shows training compute (Eq. 7) normalized by the highest value. The vertical gray lines indicate the compute required by the baseline models. Small character-level modules tend to yield better word accuracy at the tested compute budgets.

Given a backbone size (with $L_b = H_b$ as mentioned above), we have four degrees of freedom—the number of layers $(L_e, L_d)$ and heads $(H_e, H_d)$ in encoder and decoder—which is too many for an exhaustive sweep. Therefore, we conducted some preliminary experiments, detailed in Appendix B.3, to constrain the search space. Specifically, we concluded to use the same architecture for encoder and decoder ($L_e = L_d$, $H_e = H_d$) and to fix their aspect ratio at $H_e = 2L_e$.

The crucial remaining question is how to size en- and decoder relative to the backbone. From preliminary experiments, we isolated three candidate sizes: $(H_e, L_e) \in \{(6,3), (8,4), (10,5)\}$. We then trained each candidate with multiple backbone sizes, ranging from $L_b = H_b = 16$ to $L_b = H_b = 30$. The result are depicted in Fig. 3, where we report both byte-level and word-level accuracy. Strikingly, these two metrics paint sharply different pictures, with the former favoring larger byte-level modules and the latter favoring a larger backbone. However, the relative differences in byte accuracy are very small. Moreover, byte accuracy can be improved by merely making the decoder better at completing words given the first few characters, which does not improve word accuracy.

We adopted word accuracy, which we hypothesized to be a better predictor of model quality, as our guiding metric. This lead us to choose the (6, 3) configuration for the smaller two model sizes (1B and 3B-scale). We opted to use the slightly larger (8, 4) configuration for our 7B-scale model, based on the the observed trend that larger encoder/decoder sizes catch up for larger backbones. The backbone sizes for these models are set by compute-matching the baseline, which we discuss next.

## 4.2 COMPUTE-MATCHED MODELS

For the following architecture comparison, we decided to compare compute-matched models, meaning models that require (on average) the same amount of compute to process a document from the pretraining dataset. Since compute is the primary driver of training and inference cost, we believe this approach ensures a fair comparison between architectures. We first set the sizes for the encoder and decoder modules of our hierarchical architecture based on the considerations in the previous section. For each scale, we then size the backbone of the hierarchical model to match the compute required by the baseline. For the compute matching, we used the exact computa-

Table 2: Compute-matched model configurations, showing number of attention **H**eads, **L**ayers, and **P**arameter count.

| Scale | Tokenizer Baseline | | | Hierarchical (our) | | | | | |
|---|---|---|---|---|---|---|---|---|---|
| | | | | Backbone | | | En-/decoder | | |
| | H | L | P | H | L | P | H | L | P |
| 1B | 16 | 16 | 1.1B | 18 | 18 | 1.1B | 6 | 3 | 23M |
| 3B | 24 | 24 | 3.1B | 28 | 28 | 4.3B | 6 | 3 | 24M |
| 7B | 32 | 32 | 7.0B | 36 | 36 | 9.2B | 8 | 4 | 55M |

tional cost analysis, including attention FLOPs, as explained in Appendix A.2, and the statistics of our pretraining dataset, depicted in Fig. 2. Details on the compute matching methodology may be

Table 1: Results of our pretraining experiments, showing accuracy on the pretraining dataset as well as scores on established eval tasks in the zero-shot setting. Generally, our hierarchical model performs on par with the tokenizer baseline within each compute-matched scale. There are some notable wins for the hierarchical model on the Lambada (LBD) eval task, where it outperforms the baseline by a relative margin of up to 68% (at the 7B scale).

| Model | DCLM Word Acc | DCLM Byte Acc | MMLU | LBD OAI | LBD OAI C | LBD | LBD C | ARC C | ARC E | OpenBook QA | TriviaQA | TFQA | WinoGr | HellaSwag | WiC | WebQs | PIQA | BoolQ | XNLI |
|---|---|---|---|---|---|---|---|---|---|---|---|---|---|---|---|---|---|---|---|
| **1B** | | | | | | | | | | | | | | | | | | | |
| Hierarchical (whitespace) | **35.5** | **83.7** | 26.0 | **64.6** | **8.8** | **56.5** | **3.3** | 30.5 | 65.0 | 24.8 | 9.6 | 29.1 | **61.4** | **46.5** | **54.7** | 4.0 | 73.0 | 60.2 | **37.5** |
| Baseline | 35.3 | - | **27.6** | 57.7 | 3.6 | 53.7 | 0.8 | **31.7** | **68.0** | **25.8** | **15.3** | 26.8 | 60.1 | 46.2 | 50.5 | **6.4** | **75.9** | **62.6** | 33.9 |
| MegaByte (8-byte split) | - | 80.0 | 25.1 | 51.3 | 2.1 | 41.0 | 0.7 | 23.4 | 54.9 | 20.2 | 2.9 | **30.7** | 53.0 | 38.9 | 52.0 | 1.5 | 68.8 | 55.8 | 34.4 |
| Hierarchical (8-byte split) | - | 82.6 | 25.9 | 57.3 | 2.1 | 48.1 | 0.6 | 28.2 | 62.8 | 21.0 | 5.1 | 29.1 | 56.9 | 43.8 | 48.1 | 2.6 | 69.6 | 58.2 | 34.1 |
| **3B** | | | | | | | | | | | | | | | | | | | |
| Hierarchical (whitespace) | **37.8** | - | 28.7 | **70.7** | **28.4** | **64.2** | **18.0** | 37.1 | 72.3 | 29.0 | **24.2** | **29.9** | 64.1 | 53.2 | **51.1** | **9.6** | 75.4 | **69.8** | **34.5** |
| Baseline | 37.7 | - | **29.1** | 65.8 | 19.3 | 62.1 | 15.1 | **39.8** | **72.9** | **30.0** | 24.0 | 29.1 | **65.7** | **53.3** | 50.0 | 7.3 | **77.5** | 69.7 | 34.1 |
| **7B** | | | | | | | | | | | | | | | | | | | |
| Hierarchical (whitespace) | 39.0 | - | 32.0 | **73.8** | **43.1** | **67.4** | **28.6** | 44.2 | 76.6 | 30.8 | **33.1** | **31.5** | **69.0** | 56.3 | 47.3 | **11.8** | 77.7 | 72.0 | **40.5** |
| Baseline | **39.2** | - | **32.6** | 67.9 | 25.6 | 66.0 | 24.4 | **46.8** | **79.4** | **33.4** | 32.4 | 28.6 | 68.5 | **57.3** | **49.1** | 9.4 | **79.6** | **73.3** | 37.4 |

found in Appendix A.3. The resulting models are listed in Table 2. Thanks to an efficient batched implementation of the character-level models using flash attention (Dao et al., 2022), we also see comparable step durations for our compute matched models (see Appendix A.3).

## 4.3 PRETRAINING RESULTS

In this section, we compare our hierarchical architecture with a subword tokenizer baseline for pretraining on the DCLM-Baseline dataset. All models are trained *from scratch*. We compare compute-matched models at three scales, corresponding to 1B, 3B, and 7B baselines. Table 1 presents byte- and word-level accuracy, as well as zero-shot performance across 17 standard downstream tasks. Across scales and evaluation tasks, both architectures perform similarly, with a few notable differences. At the 1B scale, the tokenizer-based model holds a modest advantage on TriviaQA, though this gap disappears at larger scales. On the other hand, the hierarchical model consistently outperforms the baseline on the Lambada evaluation, with a relative margin of up to 68%.

**Comparison to MegaByte** At the 1B scale, we also conducted a comparison with MegaByte (Yu et al., 2023), another hierarchical byte-level generative architecture, as detailed in Section 3. We use a model configuration from the original paper, which is compute-matched with our 1B-scale (14 layers in the backbone with hidden dimension $D = 2048$, 18 layers in the decoder with hidden size $d = 1024$ and learning rate $2 \cdot 10^{-4}$). The results in Table 1 show that MegaByte underperforms across all but one evaluation tasks. To further investigate, we conducted an ablation combining our hierarchical architecture with MegaByte's fixed 8-byte splitting, labeled as *Hierarchical (8-byte split)* in the table. This variant significantly improves over MegaByte (e.g., a 2.6 ppt increase in byte accuracy), showing that our hierarchical architecture is more performant even when using the same splitting rule. However, the hierarchical architecture with whitespace splitting still comes out on top, suggesting that a semantically meaningful splitting is a valuable inductive bias for the model.

## 4.4 ROBUSTNESS AGAINST INPUT PERTURBATIONS

Next, we evaluate the robustness of hierarchical and baseline models against perturbations of the inputs. We conduct this experiment on a subset of five eval tasks, for which the two architectures showed similar performance. We apply perturbations to the prompt of each item of the dataset and measure the change in average accuracy compared to each model's performance on the original (unperturbed) golden answer. The perturbations include permuting, randomizing, or deleting 10% of the characters per word, as well as changing the prompt to all caps. The results are depicted in Figure 4. We see that the hierarchical model is significantly more robust than the tokenizer-based

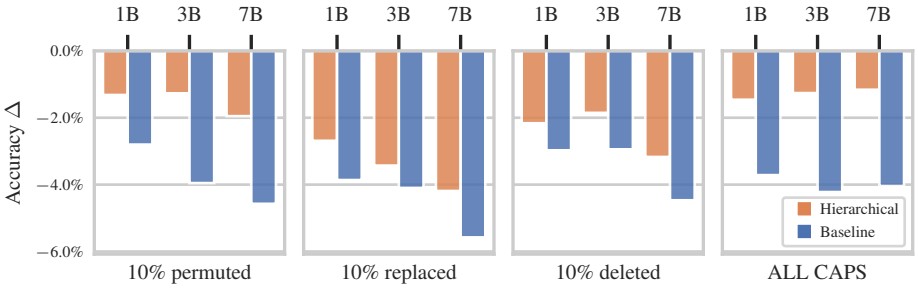

Figure 4: Average change in accuracy across the MMLU, OpenBookQA, Arc Challenge and HellaSwag eval tasks for perturbations to the prompt.

model across perturbation types and scales. In particular for the all-caps perturbation, the baseline model suffers a 3 times larger drop in accuracy. Individual results and more details may be found in Appendix B.5. Table 3 shows some illustrative examples, comparing the two models' completions to perturbed prompts.

Table 3: Example completions of perturbed prompts.

| | Prompt | Hierarchical (our) | Baseline |
|---|---|---|---|
| Incomplete words | The quick brown fox jumps over the la | zy dog. | la la la la la [...] |
| | Unce upon a ti | me, there was [...] | ime, there was [...] |
| Spelling variations | The quick brown fox | jumps over the lazy dog | jumps over the lazy dog |
| | THE QUICK BROWN FOX | JUMPED OVER THE LAZY DOG | by John Updike |
| | The quick brwon | fox jumps over the lazy dog. | -out of the 1980s, [...] |

## 4.5 ADAPTATION ON CROSS-LINGUAL CONTINUED PRETRAINING

After pretraining on the English-only DCLM-Baseline dataset, we continue training the 3B-scale models on the German Occiglot dataset (Brack et al., 2024) to test adaption to a shift in data distribution. We re-warm the learning rate to half of its initial value and train for 20k steps with otherwise identical settings as the pretraining runs. We conducted downstream evaluations every 2000 steps on a set of tasks for which both English and German versions are available, see Appendix B.6. The results are shown in Figure 5. The hierarchical model consistently achieves higher average accuracy on the German evaluations while also retaining better scores on the English tasks. Since the tokenizer operates on a rigid vocabulary, it is unable to adapt to the new domain and must resort to byte fallback or combine tokens in statistically unfounded ways. We attribute the performance difference to the resulting larger distribution shift in the input of the tokenizer based model.

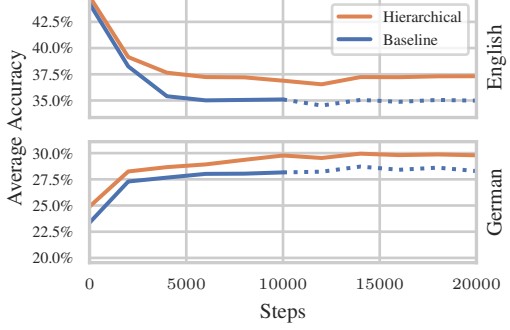

Figure 5: Continued pretraining on Occiglot German. We show average eval scores on English and German. The hierarchical architecture adapts better to the new language, while also retaining higher scores on English evals. At an equal FLOP budget we are only able to train the baseline for half as many steps due to it requiring almost two times the compute per document on the german dataset. Nevertheless, we continued training the baseline for more steps, depicted as dotted lines.

It is important to note that the models compared here have been compute-matched based on the statistics of the English DCLM dataset. As shown in Figure 2 (right), the word and token statistics shift drastically when switching to the German dataset, with bytes-per-token decreasing substantially due to tokenizer fragmentation. Consequently, the token sequence length grows much larger than

the word sequence length, making the continued pretraining of the hierarchical model **1.9 times faster** than the baseline model, or equivalently being able to train on almost twice the data for the hierarchical model.

## 5 CONCLUSION

We presented a hierarchical autoregressive transformer architecture that integrates character-level and word-level processing. Our approach retains the sequence length compression of word-level tokenization, while removing the need for a rigid, predefined vocabulary. Through extensive experiments, including models scaled up to 7 billion parameters, we demonstrated that the hierarchical architecture matches the downstream task performance of computed-matched tokenizer-based models, while significantly improving robustness to input perturbations and continued pretraining on out-of-distribution data, such as previously undersampled languages. These findings highlight the potential of hierarchical transformers to enhance flexibility, robustness, and generalization across diverse NLP tasks and domains.

**Limitations.** The whitespace splitting we used above is tailored to alphabetic languages rather than logographic languages like Chinese, where characters represent entire words or morphemes. These languages may benefit from a custom splitting rule to group bytes into semantically meaningful units. A similar concern holds for domains like mathematical writing or code, where whitespace splitting might not yield an optimal chunking. In follow-up experiments, presented in Appendix C, we obtained promising results using a "universal" splitter based on the Unicode standard. Secondly, as discussed above, for a compute-matched model, the hierarchical architecture will have a higher parameter footprint. During inference, this may be offset by a reduced size of the KV cache.

**Outlook.** Our work can be extended in various ways. First, one could experiment with different models for encoder and decoder. For example, the small character vocabulary, may facilitate multi-token prediction (Gloeckle et al., 2024) with multiple output heads. Further, with few characters per word, a text diffusion model (e.g., Li et al., 2022) could be used as a decoder. Finally, one could investigate additional levels of hierarchy, such as sentences or paragraphs, which may improve long context generation abilities.

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

## A  MODEL DETAILS

### A.1  ARCHITECTURE DETAILS

**Splitting Rule**  We split at whitespace characters, as per the Unicode standard, which includes spaces, tabs, newlines, et cetera. All consecutive whitespace characters are appended to the previous word. Our approach is agnostic to the type of splitting rule. For instance, in Section 4, we experiment with a fixed-size splitting, as in Yu et al. (2023). Other splitting rules me be adequate for non-alphabetic languages or domains like mathematical writing or code. We leave this to future work.

**End of Document**  We append a final dummy word to the end of each document to indicate its end. This dummy word consists of a single special token [S] and, like every word, is prepended with a [W] token during processing. We treat [S] as a termination token during inference. During training, we omit the final [W] as prediction target for that word, since [S] already indicates termination.

### A.2  EXACT COMPUTATIONAL COST COMPARISON

We give a detailed description of the computational cost, extending the simplified analysis presented in Section 2.3. Following common practice, we count the number of multiplications in the operations involved in a forward pass through the model, covering only matrix multiplications and ignoring biases, normalization layers, activation functions and other minor operations.

Consider a forward pass through a single transformer layer with hidden size $D$ and sequence length $S$. In a standard Llama architecture (Touvron et al., 2023), this layer has $12D^2$ parameters. This includes key, query, value and attention dense-out weight matrices of shape $D \times D$ and three matrices of shape $D \times \frac{8}{3}D$ for the SwiGLU-MLP. The matrix multiplications in the forward pass require one multiplication for each weight,

$$C_{\text{ff}}(S, D) = 12SD^2. \tag{8}$$

Additionally, we now factor in the multiplication of the query and key matrix, as well as the multiplication of the attention matrix with the value matrix, resulting in an additional cost of

$$C_{\text{attn}}(S, D) = 2S^2D. \tag{9}$$

Now consider a document consisting of $S$ characters, which get split into $S_{\text{T}}$ tokens and $S_{\text{W}}$ words, respectively. For simplicity, we assume that all words are of the same length $S/S_{\text{W}}$.

**Baseline**  The baseline model processes a sequence of length $S_{\text{T}}$, passing it through the backbone and the output head, incurring a total cost of

$$
\begin{aligned}
C_{\text{baseline}} = \quad & L_{\text{baseline}}^{\text{backbone}} C_{\text{ff}}(S_{\text{T}}, D_{\text{baseline}}^{\text{backbone}}) && \text{(Feed-forward)} \\
& + L_{\text{baseline}}^{\text{backbone}} C_{\text{attn}}(S_{\text{T}}, D_{\text{baseline}}^{\text{backbone}}) && \text{(Attention)} \\
& + S_{\text{T}} D_{\text{baseline}}^{\text{backbone}} V_{\text{T}} && \text{(LM Head)}
\end{aligned}
\tag{10}
$$

**Hierarchical Architecture**  In the hierarchical architecture, the backbone processes a sequence of length $S_{\text{W}}$. The two byte-level models process each word in the document, each of which is of length $1 + S/S_W$ due to the appended word separator tokens. Additionally, we have the two linear projections at the intersection of backbone and encoder/decoder, each of which incur cost of $D_{\text{hierarchical}}^{\text{backbone}} D_{\text{hierarchical}}^{\text{char}}$ per character. Finally, the encoder has a character-level LM head, which adds cost of $D_{\text{hierarchical}}^{\text{char}} V_{\text{B}}$. In total

$$
\begin{aligned}
C_{\text{hierarchical}} = \quad & L_{\text{hierarchical}}^{\text{backbone}} C_{\text{ff}}(S_{\text{W}}, D_{\text{hierarchical}}^{\text{backbone}}) && \text{(Backbone feed-forward)} \\
& + L_{\text{hierarchical}}^{\text{backbone}} C_{\text{attn}}(S_{\text{W}}, D_{\text{hierarchical}}^{\text{backbone}}) && \text{(Backbone attention)} \\
& + 2L_{\text{hierarchical}}^{\text{char}} S_{\text{W}} C_{\text{ff}}(1 + S/S_{\text{W}}, D_{\text{hierarchical}}^{\text{char}}) && \text{(En/decoder feed-forward)} \\
& + 2L_{\text{hierarchical}}^{\text{char}} S_{\text{W}} C_{\text{attn}}(1 + S/S_{\text{W}}, D_{\text{hierarchical}}^{\text{char}}) && \text{(En/decoder attention)} \\
& + 2S_{\text{W}} D_{\text{hierarchical}}^{\text{backbone}} D_{\text{hierarchical}}^{\text{char}} && \text{(Linear projections)} \\
& + (S + S_{\text{W}}) D_{\text{hierarchical}}^{\text{char}} V_{\text{B}} && \text{(Decoder LM head)}
\end{aligned}
\tag{11}
$$

| Scale | Hierarchical | Baseline |
|-------|--------------|----------|
| 1B    | 0.9s         | 0.9s     |
| 3B    | 2.3s         | 1.6s     |
| 7B    | 4.5s         | 4.7s     |

Table 4: Average step durations on 256 H100s at a global batch size of 1024 sequences with 16384 bytes of data

### A.3 COMPUTE MATCHING

We randomly sample $10,000$ documents from our pretraining dataset. For a given model configuration, be it a baseline model or a hierarchical model, we can now approximate its average cost by averaging the cost formulae (Eq. 10 or 11, respectively) over our sample of documents. The cost only depends on the document lengths in characters, tokens, and words.

The baseline model parameters are fixed, so we can compute its average cost. For the hierarchical model, encoder and decoder size are fixed and the only variable parameter is the number of heads and layers in the backbone. We compute average cost for possible backbone sizes and choose the size whose average cost matches that of the baseline model as closely as possible. The resulting hierarchical model configurations are listed in Table 2. Since the number of backbone heads/layers is an integer quantity, the matching is not exact, but the relative deviation is smaller than 5% across all configurations.

The average step durations for the models during pretraining on 256 H100s are shown in Table 4.

### A.4 INFERENCE-TIME PERFORMANCE

We briefly discuss the inference-time performance of our hierarchical model compared to a compute-matched baseline model. We assume we use KV caching and ignore attention FLOPs. Then the average FLOPs required to generate $S$ bytes is matched

$$S_{\mathrm{T}} C_{\mathrm{baseline}} \approx S C_{\mathrm{hierarchical}}^{\mathrm{encoder}} + S_{\mathrm{W}} C_{\mathrm{hierarchical}}^{\mathrm{backbone}} + S C_{\mathrm{hierarchical}}^{\mathrm{decoder}}. \tag{12}$$

Wall-clock time is proportional to cost if all architectures process single tokens, i.e.,

$$S_{\mathrm{T}} t_{\mathrm{baseline}} \approx S t_{\mathrm{hierarchical}}^{\mathrm{encoder}} + S_{\mathrm{W}} t_{\mathrm{hierarchical}}^{\mathrm{backbone}} + S t_{\mathrm{hierarchical}}^{\mathrm{decoder}}. \tag{13}$$

However, in an inference setting, the hierarchical architecture has the advantage that the *encoder* will always ever be queried with entire words, not single characters. Since the wall-clock time of forward pass in KV-cached inference is typically dominated by I/O operations, the encoder will take a similar time to process an entire word as it would to process a single character. Hence, in terms of wall-clock time, we will have

$$S_{\mathrm{T}} t_{\mathrm{baseline}} \quad \mathrm{vs} \quad S_{\mathrm{W}} (t_{\mathrm{hierarchical}}^{\mathrm{encoder}} + t_{\mathrm{hierarchical}}^{\mathrm{backbone}}) + S t_{\mathrm{hierarchical}}^{\mathrm{decoder}}. \tag{14}$$

This advantage could be further expanded with a simple inference-time performance optimisation of our hierarchical model. After training, one could extract and store the word embeddings (i.e., encoder outputs) for the $V_{\mathrm{W}}$ most common words in a reference corpus. One could probably cover $\geq 95\%$ of words with a very modest vocabulary size. For words in this vocabulary, the encoder stage is then replaced with an $\mathcal{O}(1)$ lookup. Likewise, for the decoder, one could store predictive word embeddings for the words in the corpus. This match could be used for speculative decoding or even be accepted as is if the match is "good enough". We leave the implementation and evaluation of this performance optimisation to future work.

### A.5 INFERENCE-TIME MEMORY

The size of the KV cache is proportional to $SLD$, where $S$ is the sequence length, $L$ is the number of layers and $D$ is the hidden dimension. For the hierarchical model, one would only cache activations in the backbone in any practical setting. Encoder and decoder operate on very small sequence lengths at inference time, where KV caching would not yield wall-clock time speedups, see also Appendix A.4. Hence, we compare $S_{\mathrm{T}} L_{\mathrm{baseline}}^{\mathrm{backbone}} D_{\mathrm{baseline}}^{\mathrm{backbone}}$ to $S_{\mathrm{W}} L_{\mathrm{hierarchical}}^{\mathrm{backbone}} D_{\mathrm{hierarchical}}^{\mathrm{backbone}}$. For

our compute-matched models, using $S_T$ and $S_W$ from Figure 2, the size of the KV cache would be reduced by 6-13%, depending on the exact configuration. Note that this reduction would be far more pronounced for out-of-domain inference, where $S_T$ und $S_W$ can shift drastically.

## B  EXPERIMENT DETAILS

### B.1  TRAINING SETTINGS

Following Dubey et al. (2024), the peak learning rate is set to $\mathrm{lr}_{32} = 3 \cdot 10^{-4}$ for the 32 head model and scaled with model size in terms of number of heads as $\mathrm{lr}(H) = \frac{32}{H}\mathrm{lr}_{32}$. For the hierarchical model, we use the number of heads in the backbone; we briefly verified that the heuristic is adequate for the hierarchical architecture. Since it is not tailored to a hierarchical model, there might be room for further improvement.

### B.2  WORD-LEVEL ACCURACY

For a given input sequence $x_{1:T}$ and a segment $x_{1:t}$, we denote the model's prediction as $m(x_{1:t}) \in [0,1]^{|\mathbb{B}|}$, which is a vector of predictive probabilities for the next byte $x_{t+1}$. Byte-level accuracy is

$$A_{\text{byte}}(x_{1:T}) = \frac{1}{T-1} \sum_{t=1}^{T-1} \delta\left( m(x_{1:t})_{x_{t+1}} = \max_i m(x_{1:t})_i \right). \tag{15}$$

Now assume the byte sequence is segmented into words, given by a set of indices $s_1, \ldots s_W \in [T]$ indicating the *first* character of a word. The next-byte probabilities auto-regressively imply next-word probabilities, for which we can compute a word-level accuracy as

$$A_{\text{word}}(x_{1:T}, s_{1:W}) = \frac{1}{W-1} \sum_{w=1}^{W-1} \prod_{t=s_w}^{s_{w+1}-2} \delta\left( m(x_{1:t})_{x_{t+1}} = \max_i m(x_{1:t})_i \right). \tag{16}$$

### B.3  HIERARCHICAL ARCHITECTURE SWEEP

**Aspect Ratio**  We first ran an experiment concerning the heads:layers aspect ratio in encoder and decoder, wanting to understand whether it may be beneficial to deviate from the 1:1 ratio used in a standard Llama architecture. For this experiment, we tied the sizes of encoder and decoder ($L_d = L_e$, $H_d = H_e$) and tested different values for the aspect ratio. We hypothesized that a "wider" model may be beneficial and tested aspect ratios 1:1, 3:2, and 2:1. The result is depicted in Figure 6. Since the result didn't show any considerable difference between different aspect ratios, we went with 2:1 based on the intuition that the change in hidden size between character-level and word-level modules should be limited when scaling to larger backbones.

**Encoder/Decoder Balance**  Next, we ran an experiment to decide how encoder and decoder should be sized relative to each other. We used $H_e = H_d = 8$ and allocated a total number of $L_e + L_d = 8$ layers to be distributed between encoder and decoder. As shown in Figure 7, the best word accuracy is achieved by an even number of layers in the two modules. Similar as in Figure 3, byte-level accuracy favors a larger decoder module.

### B.4  EVALUATION TASKS

We use a set of established downstream evaluation tasks, implemented in evaluation suites like the Eleuther AI eval harness (Gao et al., 2024). In the following, we give brief descriptions as well as citations. The descriptions are quotes either from the original paper or from Li et al. (2024).

- **MMLU** (Hendrycks et al., 2021) is a 4-way multiple choice question answering dataset that covers 57 different domains and tasks, evaluating both world knowledge and problem solving capabilities.

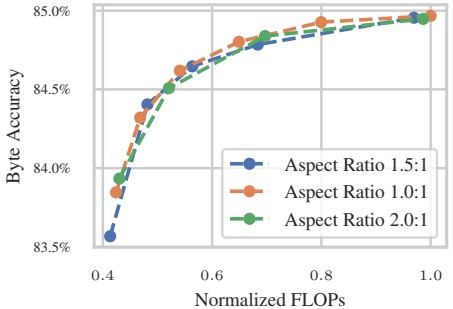
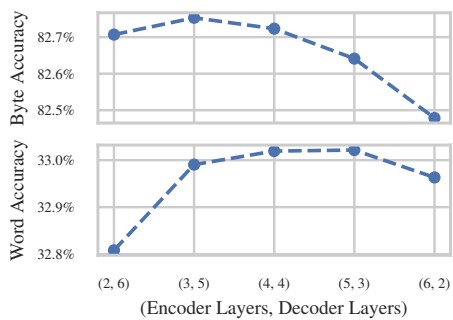

Figure 6: Scale up of the character-level module size at different aspect ratios and a fixed backbone size of 16 heads and layers. The x-axis depicts compute cost (Eq. 7), normalized by the largest value.

Figure 7: Encoder-decoder balance. A fixed number of 8 character-level layers is distributed between encoder and decoder. The backbone is fixed.

- **LBD:** LAMBADA (Paperno et al., 2016) is a collection of narratives where a human is able to guess the final word of the narrative, but is not able to if they are only given the final sentence. To perform well on this task requires the model to attend to context from the full narrative and cannot simply rely on the local context.

- **ARC:** The ARC easy and ARC challenge datasets (Clark et al., 2018) contain four-way multiple choice questions taken from grade 3-9 science exams, where questions in the easy dataset require knowledge of basic science, and the challenge questions require some procedural reasoning.

- **OpenBook QA** (Mihaylov et al., 2018) is a 4-way multiple choice question answering dataset that requires the model to use multi-step reasoning and commonsense knowledge.

- **TriviaQA** (Joshi et al., 2017) is an open-ended question answering dataset that evaluates the world knowledge of a model.

- **TFQA:** TruthfulQA (Lin et al., 2022) is a benchmark to measure whether a language model is truthful in generating answers to questions. The benchmark comprises 817 questions that span 38 categories, including health, law, finance and politics. Questions are crafted so that some humans would answer falsely due to a false belief or misconception. To perform well, models must avoid generating false answers learned from imitating human texts.

- **WinoGr:** The Winogrande dataset (Sakaguchi et al., 2021) extends the Winograd Schema Challenge dataset by expanding the dataset to a wider variety of domains.

- **HellaSwag** (Zellers et al., 2019) is a 4-way multiple choice commonsense reasoning dataset, where the model is required to understand implicit context and common knowledge in order to correctly select the continuation to a context.

- **WiC:** Word in Context (Pilehvar & Camacho-Collados, 2019). WiC is a benchmark for the evaluation of context-sensitive word embeddings. WiC is framed as a binary classification task. Each instance in WiC has a target word w, either a verb or a noun, for which two contexts are provided. Each of these contexts triggers a specific meaning of w. The task is to identify if the occurrences of w in the two contexts correspond to the same meaning or not. In fact, the dataset can also be viewed as an application of Word Sense Disambiguation in practise.

- **WebQs:** The Web Questions dataset (Berant et al., 2013) consists of 6,642 question/answer pairs. The questions are supposed to be answerable by Freebase, a large knowledge graph. The questions are mostly centered around a single named entity. The questions are popular ones asked on the web (at least in 2013).

- **PIQA** (Bisk et al., 2019) is a binary multiple choice question answering dataset that requires the model to use physical commonsense reasoning to answer correctly.

- **BoolQ** (Clark et al., 2019) is a binary question answering dataset where the model is expected to answer questions about relevant passages.

- **XNLI** (Conneau et al., 2018) is a subset of a few thousand examples from MNLI which has been translated into a 14 different languages (some low-ish resource). As with MNLI, the goal is to predict textual entailment (does sentence A imply/contradict/neither sentence B) and is a classification task (given two sentences, predict one of three labels).

## B.5 DETAILS ON ROBUSTNESS EVALUATIONS

Figure 8 shows the complete robustness results, initially reported in Section 4.4, displayed separately for each eval task.

## B.6 DETAILS ON CONTINUED PRETRAINING EXPERIMENT

We average over the following benchmarks, for which we have English and German versions:

- HellaSwag, machine translated (Zellers et al., 2019; Plüster, 2024)
- Arc Challenge, machine translated (Clark et al., 2018; Plüster, 2024)
- Truthfulqa, machine translated (Lin et al., 2022; Plüster, 2024)
- Lambada OpenAI, machine translated (Gao et al., 2024)
- MMLU, Human translated (MMMLU) (Hendrycks et al., 2021; OpenAI, 2024)

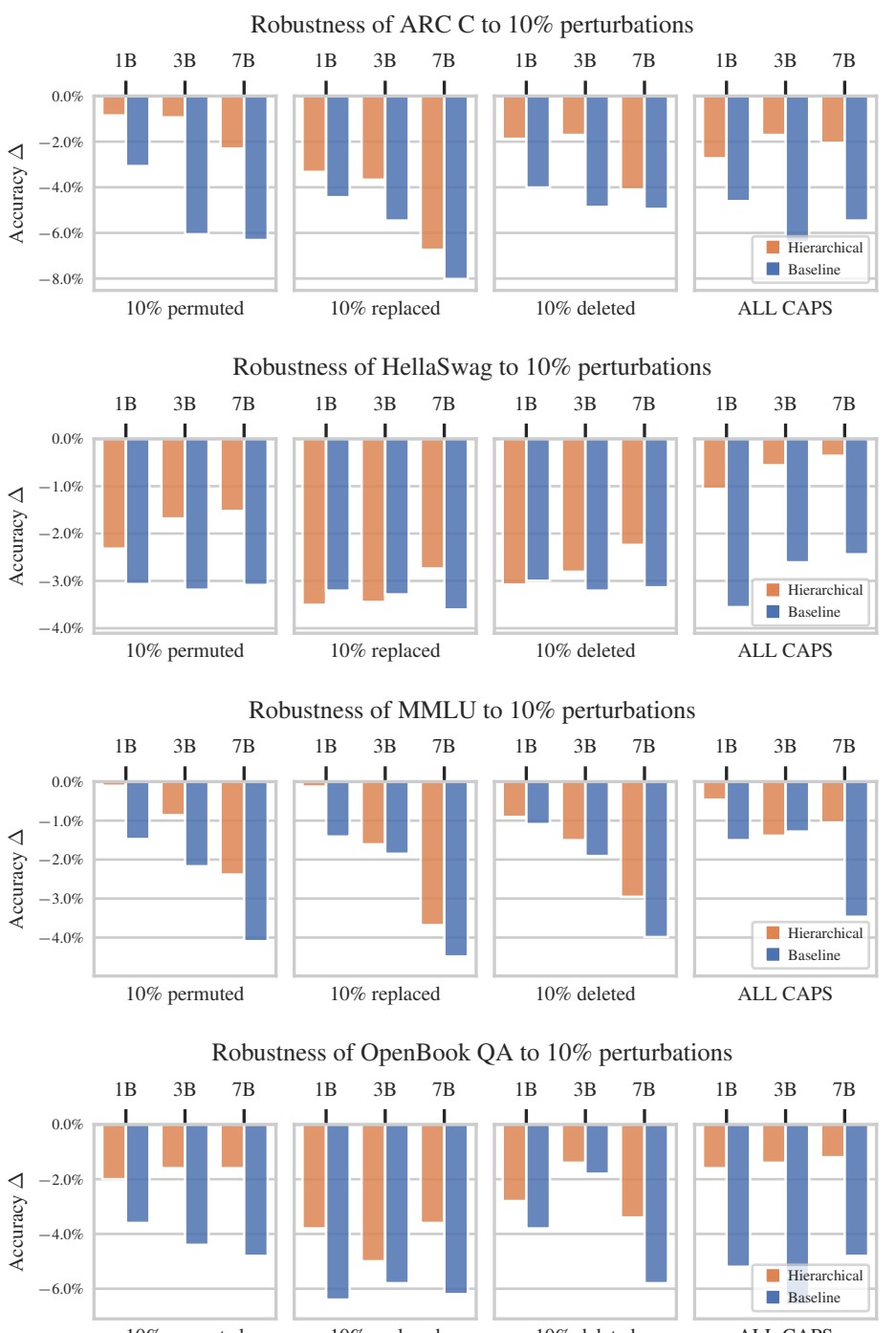

Figure 8: Robustness results.

Table 5: Results of pretraining experiments with the Unicode splitter, showing scores on established eval tasks in the zero-shot setting. The Unicode splitter outperforms the whitespace splitter on the majority of eval tasks.

| Model | MMLU | LBD OAI | LBD OAI C | LBD | LBD C | ARC C | ARC E | OpenBook QA | TriviaQA | TFQA | WinoGr | HellaSwag | WiC | WebQs | PIQA | BoolQ | XNLI |
|---|---|---|---|---|---|---|---|---|---|---|---|---|---|---|---|---|---|
| **3B** | | | | | | | | | | | | | | | | | |
| Hierarchical (Unicode) | **31.0** | **70.9** | **29.5** | **65.7** | **18.3** | 39.0 | **73.4** | 27.8 | **25.9** | 27.8 | **65.9** | 52.8 | 50.0 | **11.0** | 76.8 | 69.1 | **37.7** |
| Hierarchical (Whitespace) | 28.7 | 70.7 | 28.4 | 64.2 | 18.0 | 37.1 | 72.3 | 29.0 | 24.2 | **29.9** | 64.1 | 53.2 | **51.1** | 9.6 | 75.4 | **69.8** | 34.5 |
| Baseline | 29.1 | 65.8 | 19.3 | 62.1 | 15.1 | **39.8** | 72.9 | **30.0** | 24.0 | 29.1 | 65.7 | **53.3** | 50.0 | 7.3 | **77.5** | 69.7 | 34.1 |

## C  UNICODE SPLITTER

In our main experiments, we used a simple whitespace splitting rule, which showed competitive performance on natural text in english and german. As discussed in Section 5, whitespace splitting is not suited for languages without explicit word separators (Chinese, Japanes, Korean) and code, where punctuation is used as a separator. To address these limitations we experimented with alternative split rules and found a promising universal rule in the Unicode standard for word boundaries, which we detail and evaluate in this section.

### C.1  SPLIT RULE

The Unicode Standard, in particular Unicode Standard Annex #29, provides "guidelines for determining default segmentation boundaries between certain significant text elements", which includes word segmentation. This also covers word boundaries for non-alphabetic languages. We use the `uniseg` Python package[3] to split text into words according to this standard. In addition, we have found it to be beneficial to split text at punctuation. To improve sequence compression, we merge leading whitespaces and trailing punctuation into words. In the following, we refer to this splitting rule as the Unicode splitter.

### C.2  PRETRAINING RESULTS

We repeat our DCLM-Baseline pretraining experiments using the Unicode splitter. All training details stay as described in Section 4. The byte-per-word statistics of the new splitter result in the same flop-matched model sizes described in Section 4.2. The results are shown in Table 5, where we can see that the Unicode splitter outperforms the whitespace splitter on the majority of eval tasks.

### C.3  CROSS-LINGUAL CONTINUED PRETRAINING ON CHINESE DATA

Next, to test cross-lingual adaptation to a non-alphabetic language, we perform a continued pretraining on the Skypile dataset (Wei et al., 2023), which is a Chinese pretraining dataset. This experiment has been done using the 3B model scale using 5k steps; all other experimental details match those described in the German experiment presented in Section 4.5. We compare the hierarchical architecture using the Unicode splitter to our tokenizer baseline.

Since we did not have access to Chinese-language downstream evals, we report *bits per byte (bpb)*[4]. The learning curves are depicted in Figure 9 and the final bits per byte on a held-out portion of SykPile is shown in Table 6. We see that the hierarchical architecture reduces bpb more rapidly and reaches significantly better values within the assigned training step budget.

---

[3]https://uniseg-py.readthedocs.io/en/latest/wordbreak.html

[4]For an input document $x$ we define $\text{BPB}(x) := -\frac{1}{N_b} \sum_{i=1}^{N} \log_2 P(x_i | x_{<i})$, where $N_b$ is the number of UTF-8 bytes in the text, $N$ is the number of elements processed by the model (Equal to $N_b$ for the hierarchical model and equal to the number of tokens in the tokenizer based model) and $P$ is the next item probability as predicted by the model.

Table 6: Bits per byte on a held-out portion of SkyPile after after continued pretraining.

| Model | Bits per Byte |
|---|---|
| Tokenizer Baseline | 0.94 |
| Hierarchical (Unicode) | 0.80 |
| Hierarchical (Whitespace → Unicode) | 0.80 |

As in the corresponding experiment using German data, the tokenizer baseline also needs substantially more compute for the same number of training steps, due to tokenizer fragmentation. Since the tokenizer is not attuned to Chinese, it almost always goes into a byte fallback, degrading to an average bytes-per-token of just 1.02, compared to 4.29 bytes-per-word for Unicode splitter. Overall, the tokenizer version incurs 2.3 times the computational cost.

Fig. 10 shows a variant of the previous experiment, where we exchange the splitting rule after the initial (English-only) pretraining experiment. That is, the learning curve labeled *Whitespace →  Unicode* has been pretrained on DCLM using the whitespace splitter and fine-tuned on Skypile using the Unicode splitter. For reference, we also display the learning curve of the previous experiment, which uses the Unicode splitter throughout. We can see that the model adapts to the change in splitting rule rapidly, catching up with the "continuous" run within approximately 1000 training steps. This rapid adaptation suggests that the backbone learns language-agnostic representations, while the encoder/decoder components can effectively map new byte sequences into this established embedding space.

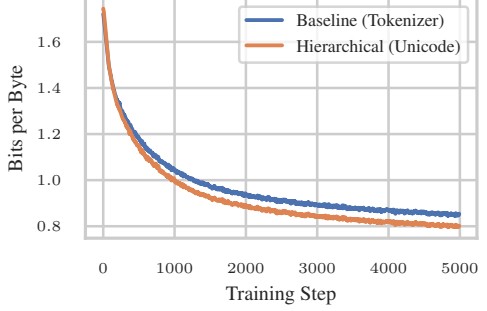

Figure 9: Learning curve in bits per byte (bpb) for continued pretraining on the Chinese Skypile dataset. Using the new Unicode splitter, the hierarchical architecture adapts to the new language more rapidly and achieves better bpb within the assigned step budget. Note that the baseline requires 2.3 times more compute for the same number of training steps.

Figure 10: Learning curve in bits per byte (bpb) for continued pretraining on the Chinese Skypile dataset. Both variants use the Unicode splitter for the continued pretraining stage, but have been trained using different splitters in the initial English-only pretraining phase. The variant that undergoes a change of splitting rule adapts quickly and achieves the same final performance.

### C.4  INFERENCE PERFORMANCE ON OUT-OF-DISTRIBUTION DATA

We also applied the models pretrained on DCLM to out-of-distribution text without any additional training. We use the GitHub Code[5], OpenWebMath (Paster et al., 2023), Pile of Law (Henderson et al., 2022), and SkyPile datasets. We report bits per byte in Table 7. Overall, there are no significant differences between the hierarchial and the tokenizer-based model. However, we also computed inference FLOPs on these out-of-distribution datasets, shown in Table 8. As in our continued pretraining experiments, we see a significant advantage for the hierarchical model.

---

[5]https://huggingface.co/datasets/codeparrot/github-code

Table 7: Bits per byte (bpb) of the DCLM-pretrained models on out-of-distribution data without any additional adaptation. The two models perform on par.

| Model | GitHub Code | OpenWebMath | Pile of Law | SkyPile |
|---|---|---|---|---|
| Hierarchical (Unicode) | 0.66 | 0.81 | 0.66 | 1.74 |
| Tokenizer Baseline | 0.62 | 0.81 | 0.70 | 1.72 |

Table 8: Difference in compute required to process out-of-distribution documents, expressed as the ratio between the FLOPs required by the tokenizer baseline and the hierarchical model. The tokenizer-based model requires significantly more compute due to tokenizer fragmentation.

| Dataset | FLOP ratio Baseline/Hierarchical |
|---|---|
| Pile of Law | 1.30 |
| SkyPile | 2.32 |
| Open Web Math | 1.28 |
| GitHub Code | 1.72 |

