# OpenReview forum: "Hierarchical Autoregressive Transformers: Combining Byte- and Word-Level Processing for Robust, Adaptable Language Models"
_ICLR.cc/2025/Conference — ICLR 2025 Poster_

### Official Review · Reviewer_PbBe · 2024-11-03

**Soundness:** 1
**Presentation:** 3
**Contribution:** 2
**Rating:** 5
**Confidence:** 3

**Summary:**

The authors propose a method to avoid large lookup table in word-level tokenization by utilizing a small character-level lookup mechanism. They employ a light weight character level encoder to convert character sequences into word embeddings, which are then processed by a word-level backbone model and decoded back into characters. Compared with subword tokenization, word-level tokenization with character-level lookup reduces the lengths of sequences processed by the backbone LLM and allows a larger number of parameters under the same computation budget.

**Strengths:**

1. Compared with subword tokenization, word-level tokenization reduces the lengths of sequences processed by the backbone LLM and allows a larger number of parameters under the same computation budget.
2. Experiments show the proposed method is more robust to input perturbation and cross-lingual continued pretraining.

**Weaknesses:**

1. **I disagree with the claim that this method is tokenization-free.** Actually, it does employ a word-level tokenization approach. It heavily relies on a space-based tokenization mechanism. The backbone autoregressive model operates on word-level embeddings.
2. **The primary contribution of this work seems to be more like "how to avoid a large lookup table in word-level tokenization." The proposed method addresses this issue by utilizing a small character-level lookup table.** It aggregates the embeddings of individual characters within a word during encoding and employs an autoregressive model to decode each character from a word-level embedding during inference.
3. Clearly, this method cannot be applied to languages without using an alphabet system (e.g., Chinese, Japanese) or without a space tokenization mechanism (e.g., Arabic, Thai). Therefore, the adaptation experiments in Section 4.6 is not very convincing, which chooses languages from the same language family.
4. The contribution appears incremental compared to Megabyte [1]. **While Megabyte employs a fixed patch size, this work utilizes a predefined tokenization mechanism to segment each patch. However, this predefined tokenization approach reduces the flexibility of the method, preventing it from being considered tokenization-free.**

[1]  MEGABYTE: predicting million-byte sequences with multiscale transformers.

**Update**

**I have carefully reviewed the authors' response and acknowledge the experimental results presented in the paper. However, I think the technical content to be highly similar to previously published work [2]. The only difference lies in that [1] uses subword as a higher-level unit but this submission uses word. Both needs predefined segmentation methods. Therefore, I have decided on a final rating of 5.**

[2] Learn Your Tokens: Word-Pooled Tokenization for Language Modeling, EMNLP 2023

**Questions:**

Please see the weaknesses.

---

> ### Author Response · Authors · 2024-11-15
> **Response to initial review**
>
> Thank you for your detailed review. We appreciate that you highlighted some strengths of our approach, such as the improved sequence length compression and robustness to perturbations. Below, we address your concerns individually.
>
> > I disagree with the claim that this method is tokenization-free.
>
> We understand your point and it hinges on the ambiguity of the term tokenization. When we referred to our approach as "tokenization-free," our intention was to emphasize that we eliminate rigid, finite token-level vocabularies. While our model operates on word-level embeddings, these do not correspond to fixed vocabulary words. We still rely on a base alphabet (UTF-8 bytes) and a splitting rule, similar to pre-tokenization in BPE. We are happy to adjust the wording to reflect this distinction more clearly. Note that the term "tokenizer-free" appears only twice in the paper—once in the title and once in the introduction. We have already removed the latter. If you believe it's necessary to revise the title for clarity, we are open to that, but will need to check with the AC to confirm whether this is possible.
>
> > The primary contribution of this work seems to be more like "how to avoid a large lookup table in word-level tokenization."
>
> Avoiding a large look-up table is certainly one advantage of our approach, but it is not the complete picture. Our approach avoids reliance on a word-level vocabulary altogether. As a result, our model can, in principle, process and generate any word using the base alphabet (UTF-8 bytes). Word-level embeddings are learned as part of the end-to-end training process. We demonstrate that this approach brings tangible benefits in terms of robustness and fine-tunability, and we believe this represents a valid and significant contribution.
>
> > Clearly, this method cannot be applied to languages without using an alphabet system (e.g., Chinese, Japanese) or without a space tokenization mechanism (e.g., Arabic, Thai).
>
> Yes, we acknowledge this limitation in Section 5. However, we would like to clarify that this limitation arises from the use of whitespace splitting, not from the hierarchical auto-regressive transformer architecture itself. As mentioned at the start of Section 2, we chose whitespace splitting as one example of a splitting rule suitable for alphabetic languages, while other splitting rules may be more appropriate for different languages or domains.
>
> Since submitting the paper, we have conducted additional experiments using a splitting rule based on the Unicode definition of word boundaries (see [Unicode TR29](https://unicode.org/reports/tr29/)), which better accommodates non-alphabetic languages. The results of these experiments can be found in Appendix C of the revised paper. Notably, the Unicode splitter allows us to fine-tune a model pretrained on the English-only DCLM-Baseline dataset on a Chinese dataset (Skypile), achieving better performance than the tokenizer baseline (0.8 vs. 0.92 bits per byte) while reducing the cost per step by more than 2x. Additionally, this new splitting rule enhances performance in our English-only pretraining experiments, with an improvement of +2.3 percentage points on MMLU.
>
> > The contribution appears incremental compared to Megabyte
>
> We acknowledge that our approach is closely related to MegaByte, as discussed in our Related Work section. However, we believe our method offers several significant advantages. Notably, our model can accommodate any splitting rule, whereas MegaByte is constrained to fixed-size patches due to its reliance on concatenating character embeddings. Our experimental results clearly demonstrate that semantically meaningful splitting, such as the word-based splitting we used, offers substantial benefits. In fact, our approach outperforms MegaByte in all but one evaluation. Even in our ablation where we use the fixed-size splitting rule with our hierarchical architecture, we outperform MegaByte decisively. We have highlighted these findings more clearly in the revised version of the paper, summarizing them in a separate paragraph within Section 4.4 and adjusting the labels in Table 1.
>
> In terms of scientific contribution, we also want to emphasize that we significantly scale up the experiments compared to the MegaByte paper, demonstrating the feasibility of our approach at relevant scales. Additionally, we have thoroughly explored the robustness and fine-tunability properties of our proposed architecture.
>
> ---
>
> We hope that our response and the revised version of the paper have addressed your concerns and questions. We kindly ask you to reconsider your rating. Specifically, we are curious about the "poor" score in the "Soundness" dimension, as we believe your comments primarily relate to the "Presentation" and "Contribution" aspects of our work. Please let us know if you have any further comments or questions.

---

> > ### Comment · Reviewer_PbBe · 2024-11-26
> >
> > I have carefully reviewed the authors' rebuttal and appreciate their efforts. Based on this, I have decided to increase my score from 3 to 5.
> >
> > Here are my reasons:
> >
> > The terms "tokenization-free" and "vocabulary-free" are both misleading. The use of a split rule inherently constitutes a form of tokenization. Additionally, your methods cannot be considered "vocabulary-free" since they rely on finite discrete token sets, which effectively constitute a vocabulary. Therefore, I don't think method in this work aligns with its motivation.
> >
> > However, this work provides empirical insights into building LLMs in a hierarchical manner, which I consider a positive contribution. I also acknowledge the authors' efforts in scaling up the model through their experiments.

---

> > > ### Author Response · Authors · 2024-11-26
> > > **Let's resolve this wording issue**
> > >
> > > Dear reviewer,
> > >
> > > thank you for your consideration of our response, we appreciate your openness!
> > >
> > > It seems that your remaining concern is about the label _tokenizer-free_ or _vocabulary-free_. In that case, we would really like to work with you and resolve any remaining lack of clarity. **We believe it would be a shame to get a _"reject"_ rating over what is essentially a wording issue.**
> > >
> > > ### **What we want to express**
> > >
> > > With the attribute _tokenizer-free / vocabulary-free_, we want to express that the backbone of our proposed architecture processes text on a word level (as defined by a simple regex-based splitting rule) without the need for a rigid, predefined (sub)word-level vocabulary. It can ingest and output any word over the base alphabet, UTF-8 bytes.
> > >
> > > We believe it is fair that we want to highlight this strength of our approach. We have no intention of over-claiming or misleading readers. We did another pass over the paper and think that this is accurately reflected in the _body_ of the paper. In particular, it does not not mention the terms tokenizer-free or vocabular-free anymore (and we are in no way attached to these labels).
> > >
> > > ### **Questions for reviewer**
> > >
> > > Therefore, we would like to ask you:
> > > 1) Do you agree with our characterization above?
> > > 2) Do you think the _body_ of our paper is accurately reflecting this aspect?
> > >
> > > ### **Suggestions for title**
> > >
> > > If we can agree on the above, it should be easy to change the title to reflect this. Some suggestions:
> > >
> > > 1) Hierarchical Autoregressive Transformers: Combining Byte and Word-Level Processing for Robust, Adaptable Language Models
> > > 2) Hierarchical Autoregressive Transformers for Efficient Byte-Level Language Modelling
> > > 3) Hierarchical Autoregressive Transformers: Language Modelling Without (Sub)Word-Level Vocabularies
> > >
> > > ---
> > >
> > > Again, thank you for engaging with our response and for raising this point. We would really appreciate your help in resolving this.

---

> > > > ### Comment · Reviewer_PbBe · 2024-11-26
> > > >
> > > > I do not agree with your claim.
> > > >
> > > > 1. I want to point out that this is not just a "wording issue." I have read the responses from other reviewers, and it is clear that the claims of being "tokenization-free" or "vocabulary-free" have significantly influenced their reviews.
> > > >
> > > > 2. The method proposed in this paper has already appeared in the literature. **In fact, the approach of using character-level embedding to construct higher-level autoregressive models has been published before:**
> > > >
> > > > [1] Learn Your Tokens: Word-Pooled Tokenization for Language Modeling, EMNLP 2023
> > > >
> > > > **The only difference between [1] and this submission is that [1] uses subword as a higher-level unit but this submission uses word. Both needs predefined segmentation methods.**
> > > >
> > > > In a more generalized view, the hierarchical architecture used in this paper is very similar to the following paper:
> > > >
> > > > [2] BlockTransformer: Global-to-Local Language Modeling for Fast Inference, arXiv 2024/04
> > > > [3] SpaceByte: Towards Deleting Tokenization from Large Language Modeling, arXiv 2024/06
> > > >
> > > > However, there is no discussion about these very similar works, especially [1]. This is the major reason I don't acknowledge the novelty.
> > > >
> > > > **As I stated before, I acknowledge the experiments, especially experiments in a scaled-up settings, therefore I decided to increase my score to 5. However, considering the technical contribution, especially very high similarity to [1], I will not further increase my score.**

---

> > > > > ### Author Response · Authors · 2024-11-26
> > > > >
> > > > > Dear reviewer,
> > > > >
> > > > > we respect your decision to argue for rejecting this paper.
> > > > >
> > > > > We do, however, have to register our disagreement with your argumentation regarding our use of the term _tokenizer-free_: We have provided a clear characterization of what we wanted to highlight with this adjective. We believe the paper as a whole is absolutely clear about the precise nature of our claim and in no way misleading. Using the attribute tokenizer-free may have been a mistake, which we are absolutely willing to correct, and which should not warrant rejection. And we don't think it's a good-faith argument to suggest that the other reviewers were somehow dazzled by our use of a specific adjective.
> > > > >
> > > > > Regarding the related papers you brought up in your most recent response: We were indeed not aware of these prior works. From a quick glance, [1] in particular seems to investigate a similar architecture at toy model scale. We will read them carefully and will discuss commonalities and differences in a follow-up comment and a revised Related Work section.

---

> > > > > > ### Author Response · Authors · 2024-11-27
> > > > > > **Additional related work**
> > > > > >
> > > > > > Dear reviewer,
> > > > > >
> > > > > > we have now carefully read the papers you have brought to our attention. Thank you for pointing out these works! Paper [2] is concerned with an inference-time KV cache memory optimization, which employs a block pooling structure _on top of_ subword tokens. We therefore consider it only loosely related. Papers [1] and [3] do indeed propose similar hierarchical architectures for generative language modelling. We apologize for this oversight!
> > > > > >
> > > > > > We have added descriptions of commonalities and differences to the Related Work section. We quote the corresponding paragraphs here for reference:
> > > > > >
> > > > > > > Thawani et al. (2023) propose the hierarchical architecture most closely related to the present work. A notable difference is that they prepend not one but four [W] tokens to each word in order to increase model capacity when going from encoder to backbone. This incurs drastically higher cost in
> > > > > > en- and decoder compared to our approach of incrasing the hidden dimension. Thawani et al. (2023) experiment with models up to 77M parameters on datasets with fewer than 10M characters and a context window of only 192 characters (or the token equivalent thereof). Unfortunately, their experiments are not compute-matched and, by our calculation, assign 4x more compute to the hierarchical architecture compared to the baseline.
> > > > > > >
> > > > > > > Finally, in work that appeared concurrently with the preparation of the present paper, Slagle (2024) propose a byte-level model that applies additional Transformer layers to a subset of the input bytes. They investigate both a fixed-size spacing as well as a split rule that marks only certain bytes as “split bytes”, including whitespaces and punctuation. The byte-level layers are not restricted to individual words/chunks and instead use sliding window attention, precluding inference-time performance improvement via caching (Appendix A4). Experiments are compute-matched and scaled up to the 1B models trained on 80B bytes and do not include downstream evaluations. None of the above papers investigate robustness or finetunability
> > > > > >
> > > > > > **We have also adjusted the wording in abstract, introduction, and method section to make clear that we revisit, refine and thoroughly investigate a hierarchical architecture for generative language modelling, rather than making an entirely novel architecture proposal.**
> > > > > >
> > > > > > ---
> > > > > >
> > > > > > We acknowledge that, in light of this additional related work, the novelty on the architecture side is limited. We still believe that our paper makes a significant and valuable contribution. In particular, three out of four main contributions listed in our introduction are independent of model novelty:
> > > > > > - extensive compute- and data-matched experiments at relevant scales
> > > > > > - demonstrating robustness
> > > > > > - demonstrating improved finetunability
> > > > > >
> > > > > > With the additional experiments using the "Unicode splitter", which we added during the rebuttal, we also tackle the extension to non-alphabetic languages, which were not considered in prior work.

---

> ### Author Response · Authors · 2024-11-20
> **Friendly reminder**
>
> Dear reviewer,
>
> as we are halfway through the discussion period, we wanted to send a friendly reminder regarding our response. We believe that our response and the revised version of the paper address the points you have raised your review. If any questions or concerns persist, we would appreciate the chance to resolve them during the discussion period.
>
> Thanks and best regards!

---

> ### Author Response · Authors · 2024-11-25
> **Another reminder**
>
> Dear reviewer,
>
> the discussion period will end tomorrow. We believe we have addressed all four points that you have raised in your review. We would really appreciate to get your feedback on this before the discussion period ends. We urge you to reconsider your rating; in particular a "poor" rating on the soundness dimension seems unwarranted to us, as none of your concerns pertain to the scientific soundness of our work.
>
> Thank you!

---

> ### Comment · Area_Chair_HNnD · 2024-11-25
>
> Dear Reviewer PbBe,
>
> Thank you for your valuable contributions to the review process for the paper! The authors have submitted their rebuttal, and I would greatly appreciate it if you could take a look and provide your response.

---

### Official Review · Reviewer_KFMw · 2024-11-04

**Soundness:** 3
**Presentation:** 3
**Contribution:** 3
**Rating:** 8
**Confidence:** 4

**Summary:**

I had wrongly entered the review of another paper - I have now corrected this. I apologise and commend the authors for having tracked down the correct review!

They propose a hierarchical transformer model which does not rely on a tokeniser to preprocess the dataset into tokens of a fixed sized vocabulary. Instead it proposes an encoder decoder which takes words as a sequence of bytes and encodes and decodes the byte sequences into a word embedding which is then processed by a backbone transformer model. They show that this hierarchical model is theoretically computationally efficient in terms of FLOPs and performs well across a number of tasks. Furthermore they show that it is more robust errors than the baseline which rely on the fixed tokenisation from the original training corpus.

**Strengths:**

The model is well motivated as being a good balance between word and byte level representations.
They scale these up to 7B which other hierarchical work did not
They show their model is better than models which do not take word boundaries into account

**Weaknesses:**

They do not situate the work in the long history of hierarchical models, and character and byte based literature. Why is this paper the one that will convince us to move away from sub-word units? The case is not clearly made.
There are important details that they do not explain clearly like the dimensions of the backbone model, the test sets and their experimental design. They do not deal with the case of CKY languages.

**Questions:**

Why not discuss in more depth how this work relates to other work on byte/pixel/character based models? This is a fundamental shift in the transformer paradigm and has a lot of implications. More discussion here would have been welcome. Eg. Mielke, Sabrina J., et al. "Between words and characters: A Brief History of Open-Vocabulary Modeling and Tokenization in NLP." Computing Research Repository (arXiv) (2021).

How would your model cope with languages without whitespace characters as work separations eg. Mandarine? How would your model handle CKY languages with tiny vocabularies, or multilingual models which include CKY and many other scripts?

4.2 It was not clear what the relationship between the backbone and the encoder was. You say you keep the backbone to the 1:1 aspect ratio - what did you mean here? And if you chose the encode to be (8,4) what does this mean for the backbone/encoder? You have 8 heads and 4 layers on the encoder and decoder - but what do you have on the backbone - not clear!

In table 2 you show compute matched settings for the baseline and you model. They only match in terms of parameters for the first 1.1B model, for 3.1B your model is 4B, and for 7B your model is 9.2B which are both much bigger. Does this mean that they comparisons in Table 1 are unfair on the baseline and overestimate the advantage of your hierarchical model?

There is no explanation of the test sets in Table 1. In particular what is Lambada and why does the hierarchical model perform so well on it and what does this mean?

I like Figure 4! It clearly shows that the hierarchical model is more robust.

You do not explain the German Occiglot dataset - why is this considered out of distribution? The Llama model has likely seen quite a bit of German data.

---

> ### Author Response · Authors · 2024-11-15
> **Response to initial review [1/2]**
>
> **Note: Apparently, there has been a mix-up of the reviews between our paper and the paper titled "One Language, Many Gaps: Evaluating Dialect Fairness and Robustness of Large Language Models in Reasoning Tasks". We are responding to the review for _our_ paper and hope that the mix-up will be resolved swiftly.**
>
> Thank you for your thorough review. We are happy to hear that you appreciate the integration of byte- and word-level processing as well as the scale and results of our experimental evaluation. We will respond to your concerns and questions individually below.
>
> > They do not situate the work in the long history of hierarchical models, and character and byte based literature.
> >
> > Why not discuss in more depth how this work relates to other work on byte/pixel/character based models?
>
> You raise a valid point. While we believe that Section 3 (Related Work) adequately addresses the most closely related works, it does not fully capture the broader context. In response, we have revised this section to provide a more comprehensive overview. Specifically, we have:
> - added an introductory paragraph to position our work within the broader context,
> - cited the review paper by Mielke et al.,
> - referenced papers that augment token embeddings with character-level information, and
> - included papers proposing purely character- or byte-based models.
>
> Thank you for highlighting this oversight. We feel these additions have significantly improved our Related Work section. If you believe any specific paper or line of research is still missing, please let us know.
>
> > Why is this paper the one that will convince us to move away from sub-word units? The case is not clearly made.
>
> We don’t aspire to make the entire field move away from sub-word tokenization with a single paper ;) However, we argue that our work introduces the first generative transformer model that eliminates reliance on fixed subword tokens while matching the performance of a well-established tokenizer-based baseline at relevant scales. Furthermore, to the best of our knowledge, our paper is the first to present experimental evidence demonstrating improved robustness and fine-tunability for such an approach.
>
> > They do not deal with the case of CKY languages.
> >
> > How would your model cope with languages without whitespace characters as work separations eg. Mandarine?
>
> Yes, we acknowledge this limitation in Section 5. However, we would like to clarify that this limitation arises from the use of whitespace splitting, not from the hierarchical auto-regressive transformer architecture itself. As mentioned at the start of Section 2, we chose whitespace splitting as one example of a splitting rule suitable for alphabetic languages, while other splitting rules may be more appropriate for different languages or domains.
>
> Since submitting the paper, we have conducted additional experiments using a splitting rule based on the Unicode definition of word boundaries (see [Unicode TR29](https://unicode.org/reports/tr29/)), which better accommodates non-alphabetic languages. The results of these experiments can be found in Appendix C of the revised paper. Notably, the Unicode splitter allows us to fine-tune a model pretrained on the English-only DCLM-Baseline dataset on a Chinese dataset (Skypile), achieving better performance than the tokenizer baseline (0.8 vs. 0.92 bits per byte) while reducing the cost per step by more than 2x. Additionally, this new splitting rule enhances performance in our English-only pretraining experiments, with an improvement of +2.3 percentage points on MMLU.
>
> > How would your model handle […] multilingual models which include CKY and many other scripts?
>
> We limited our experiments to monolingual pretraining due to the availability of established, curated datasets like DCLM-Baseline. However, we believe that our end-to-end-trained approach could be particularly advantageous for multilingual pretraining, where tokenizers often require large vocabulary sizes yet still face challenges in adequately representing diverse languages (see, e.g., [1]). Systematically exploring the multilingual capabilities of our approach, however, lies beyond the scope of this paper.
>
> It is worth re-emphasizing that our method uses UTF-8 bytes as its base alphabet, allowing seamless integration of languages with different alphabets or scripts. (See also the Chinese-language experiment detailed in our previous response.)
>
>
> ***Continued in the next comment***
>
>
>
> References
>
> [1] https://arxiv.org/abs/2305.17179

---

> > ### Author Response · Authors · 2024-11-15
> > **Response to initial review [2/2]**
> >
> > > do not explain clearly like the dimensions of the backbone mode
> > >
> > > It was not clear what the relationship between the backbone and the encoder was. You say you keep the backbone to the 1:1 aspect ratio - what did you mean here? And if you chose the encode to be (8,4) what does this mean for the backbone/encoder? You have 8 heads and 4 layers on the encoder and decoder - but what do you have on the backbone - not clear!
> >
> > Thank you for pointing out a lack of clarity here.
> >
> > By aspect ratio, we refer to the ratio of the number of heads ($H$) to the number of layers ($L$) of an architecture. We realized that this may not be standard terminology and clarified this in the revised version of the paper.
> >
> > Your comment also made us realize that the dimensions of the backbone are not made sufficiently clear in Section 4.2, in particular Fig. 3, when we present our architecture sweep for encoder and decoder. We have fixed this in the revised version.
> >
> > For our experiments, we decided to fix the aspect ratio to 1:1 in the backbone, meaning $H = L$, matching a standard Llama architecture. Table 2 lists the the exact dimensions of all models used in the main experiments.
> >
> > > In table 2 you show compute matched settings for the baseline and you model. They only match in terms of parameters for the first 1.1B model, for 3.1B your model is 4B, and for 7B your model is 9.2B which are both much bigger. Does this mean that they comparisons in Table 1 are unfair on the baseline and overestimate the advantage of your hierarchical model?
> >
> > As outlined in Section 4.3, our comparisons focus on models matched in terms of compute rather than parameter count. We believe this approach is fair, as compute is the primary cost associated with training and deploying large language models (LLMs). While the community often defaults to parameter-matched comparisons, this is largely because compute scales with parameter count when models use the same or similar tokenizers. However, this assumption does not hold for our comparison, making compute-matching the more appropriate choice in this context.
> >
> > > There is no explanation of the test sets in Table 1.
> >
> > Thank you for pointing this out. The tasks in question are standard downstream evaluation benchmarks commonly included in popular evaluation suites, such as EleutherAI's [lm-evaluation-harness](https://github.com/EleutherAI/lm-evaluation-harness), and have been used in numerous prior works. However, we acknowledge that we failed to provide proper explanations and references. To address this, we have added a corresponding section in Appendix B.4.
> >
> > > what is Lambada and why does the hierarchical model perform so well on it and what does this mean?
> >
> > Lambada is a word prediction task, designed to evaluate the text understanding of a model, in particular its ability to take broader context into account. It consists of passages for which “human subjects are able to guess their last word if they are exposed to the whole passage, but not if they only see the last sentence preceding the target word.”
> >
> > We have not yet had the opportunity to conduct a deep dive into the individual evaluation results to understand why the hierarchical model performs particularly well on this task. However, we plan to investigate this in the coming days and will share our findings in a follow-up reply.
> >
> > > You do not explain the German Occiglot dataset - why is this considered out of distribution? The Llama model has likely seen quite a bit of German data.
> >
> > DCLM-Baseline is an English-only pretraining dataset. It is important to note that all models in our experiments are trained from scratch and have not been exposed to German data prior to the fine-tuning stage. This makes the fine-tuning experiment a clear example of out-of-distribution adaptation. We have clarified these points further in Section 4.6.
> >
> >  ---
> >
> > We hope that our response and the revised version of the paper have adequately addressed your concerns. In light of this, we kindly ask you to reconsider your rating, especially as you have rated our paper as "good" across all three key dimensions: soundness, presentation, and contribution. If you have any additional comments or questions, please do not hesitate to let us know.

---

> > > ### Comment · Reviewer_KFMw · 2024-11-15
> > > **Reviewer response**
> > >
> > > I appreciate your efforts in addressing my numerous criticisms. I have also read the other reviewers comments and although I agree that the term tokeniser-free is misleading and possibly even incorrect (please try change title if you can), I think that the advantage of the hierarchical approach that you present is still clear. I am happy to increase my overall score.

---

> > > > ### Author Response · Authors · 2024-11-15
> > > > **Thank you!**
> > > >
> > > > Thank you very much for your swift response and for increasing your score.
> > > >
> > > > We will propose a new title and initiate the change in a separate thread.

---

> ### Author Response · Authors · 2024-11-25
> **Lambada eval results**
>
> Dear reviewer,
>
> In the meantime, we have taken a closer look at the Lambada eval results. We have inspected a number of examples and the corresponding completions of our hierarchical model and the tokenizer-based baseline. Unfortunately, we couldn't discern a clear pattern that would explain the diverging scores.
>
> Our only (vague) hypothesis is that our hierarchical model operates at the word level and may therefore be particularly well-suited to the Lambada task, which requires the prediction of a single word given the context.

---

### Official Review · Reviewer_Kkdq · 2024-11-04

**Soundness:** 3
**Presentation:** 3
**Contribution:** 3
**Rating:** 8
**Confidence:** 3

**Summary:**

The paper proposed a technique for tokenizer-free language modeling. They design hierarchical processing to aggregate character-level embeddings into word-level embeddings. They achieved similar performance as models with word-level tokenizers while being more robust to input pertubation. They improved the continued pretraining speed when switching domain from English to German while maintaining the performance.

**Strengths:**

- A method of tokenizer-free langauge modelling that is more robust to input perturbation and input domain shift.

**Weaknesses:**

- The proposed method only works for character-based languages.

**Questions:**

- I'm curious how the model works when doing inference in the same language but when the text is from a different distrubtion (out-of-domain).

---

> ### Author Response · Authors · 2024-11-15
> **Response to initial review**
>
> Thank you for your positive review. We are glad that you highlight the improved robustness of our approach, as well as the improved speed during continued pretraining. We will respond to your concerns and questions individually below.
>
> > The proposed method only works for character-based languages.
>
> Yes, we acknowledge this limitation in Section 5. However, we would like to clarify that this limitation arises from the use of whitespace splitting, not from the hierarchical auto-regressive transformer architecture itself. As mentioned at the start of Section 2, we chose whitespace splitting as one example of a splitting rule suitable for alphabetic languages, while other splitting rules may be more appropriate for different languages or domains.
>
> Since submitting the paper, we have conducted additional experiments using a splitting rule based on the Unicode definition of word boundaries (see [Unicode TR29](https://unicode.org/reports/tr29/)), which better accommodates non-alphabetic languages. The results of these experiments can be found in Appendix C of the revised paper. Notably, the Unicode splitter allows us to fine-tune a model pretrained on the English-only DCLM-Baseline dataset on a Chinese dataset (Skypile), achieving better performance than the tokenizer baseline (0.8 vs. 0.92 bits per byte) while reducing the cost per step by more than 2x. Additionally, this new splitting rule enhances performance in our English-only pretraining experiments, with an improvement of +2.3 percentage points on MMLU.
>
> > I'm curious how the model works when doing inference in the same language but when the text is from a different distrubtion (out-of-domain)
>
> The paper already demonstrates that our approach exhibits greater robustness to certain types of distribution shifts, such as the perturbations used in our robustness evaluation (Section 4.5). Based on your review, we extended our analysis to explore out-of-distribution performance on datasets featuring code, mathematical writing, and legal texts, domains that differ significantly from the general DCLM pretraining dataset, which consists of web-scraped text. The results of this investigation are detailed in Appendix C.4 of the revised paper. In brief, we found no significant differences in predictive performance between the hierarchical model and the tokenizer baseline (see Table 7). However, processing out-of-distribution text with tokenizer-based models requires 28% to 132% more FLOPs (see Table 8). As noted in our continued pretraining experiments (Section 4.6), this increased computational cost is attributable to the substantial drop in the tokenizer’s compression rate (bytes per token) on out-of-distribution data.
>
> ----
>
> We hope that our response and the revisions to the paper have adequately addressed your two primary concerns. In light of this, we kindly ask you to reconsider your rating, especially given that you have rated our paper as "good" across all three individual dimensions: soundness, presentation, and contribution. If you have any additional comments or questions, we would be happy to engage further.

---

> ### Author Response · Authors · 2024-11-20
> **Friendly reminder**
>
> Dear reviewer,
>
> as we are halfway through the discussion period, we wanted to send a friendly reminder regarding our response. We believe that our response and the revised version of the paper address the points you have raised your review. If any questions or concerns persist, we would appreciate the chance to resolve them during the discussion period.
>
> Thanks and best regards!

---

> ### Author Response · Authors · 2024-11-25
> **Another reminder**
>
> Dear reviewer,
>
> the discussion period will end tomorrow. We believe we have addressed the two points that you have raised in your review. We would really appreciate to get your feedback on this before the discussion period ends and kindly ask you to reconsider your rating in light of our response.
>
> Thank you!

---

> ### Comment · Area_Chair_HNnD · 2024-11-25
>
> Dear Reviewer Kkdq,
>
> Thank you for your valuable contributions to the review process for the paper! The authors have submitted their rebuttal, and I would greatly appreciate it if you could take a look and provide your response.

---

> ### Comment · Reviewer_Kkdq · 2024-11-25
>
> I would like to thank the authors for their response. I think vocabulary-free is an important direction and the authors have proposed efficient algorithms, improved upon previous methods. I would maintain my score of leaning toward acceptance.

---

> > ### Author Response · Authors · 2024-11-26
> >
> > Dear reviewer,
> >
> > Thank you very much for engaging! It seems that our response has alleviated your two primary concerns. If that is the case, may we ask you to reconsider your overall rating? You have rated our work as good on all three dimensions (soundness, presentation, contribution). That is currently not reflected in the overall score.

---

### Author Response · Authors · 2024-11-25
**Title Change**

Dear AC,

Reviewer PbBe expressed the concern that assigning the attribute _tokenizer-free_ to our proposed method may be misleading, because it relies on a word splitting rule. While we don't necessarily agree, we are open to changing the title of the paper to avoid confusion.

To keep the change minimal, we would propose to simply replace _tokenizer-free_ with _vocabulary-free_.

We weren't sure whether a title change during the rebuttal phase is possible, so we didn't make the change yet. If our paper gets accepted, please advise how to proceed.

Thank you!

---

### Meta-Review · Area_Chair_HNnD · 2024-12-25

**Metareview:**

This paper proposes a hierarchical language model that avoids traditional subword tokenizers by operating at the word level while internally processing characters with a lightweight character-level encoder/decoder.


**Strengths** (1) While the idea is not new, the experimental results show clear advantages of using such a hierarchical setup, which improves modeling efficiency and robustness over perturbations; (2) The paper demonstrates improved adaptability when moving from one domain/language to another.

**Weaknesses** (1) Clear confusion in the claim. The reviewer mentioned while the paper claims "tokenizer-free", it still relies on whitespace for splitting words, and thus it is still a rule-based tokenization. Also, such a choice makes it not possible to generalize to other scripts/languages without explicit whitespace such as Chinese;
(2) Some reviewers argued the limited comparison and discussion against prior works with very similar concepts;
(3) Some reviewers raised the confusion about the parameter counts and compute matching.


**Decision**
It is a paper receiving mixed reviews especially after the rebuttal phase, making it hard to make decisions.

On one hand, I **agree** with PbBe about the claim of "tokenizer-free" are **misleading** (even the change of vocabulary-free is not correct), and the technical contributions are very limited considering its similarity with many prior works. More importantly, after checking the paper, it also seems to lack the necessary discussion of the potential inflexibility of modeling words in characters with variable lengths (e.g., efficient attention with masks).

On the other hand, all the reviewers acknowledge the efforts of the authors for the detailed experiments and the evidence shown from the benefits. All the reviewers raised their scores from the initial rating.

I am **personally leaning toward a rejection**; however, considering the champions from Kkdq and KFMw who increased their scores, I mark the case as **borderline accept**, and will leave SAC to make the final decision.

**Additional Comments On Reviewer Discussion:**

The discussions of Kkdq and KFMw mainly focused on experiments and literature, and are willing to increase the initial rating after the authors' response.
PbBe argued on the basic claim, novelty, and practical flexibility of the proposed method. While the authors tried to respond with additional explanations, and the reviewer acknowledged the experimental efforts, the discussion did not conclude with a positive rating.

---

### Decision · Program_Chairs · 2025-01-22

Accept (Poster)